

# Performance, workload, and usability in a multiscreen, multi-device, information-rich environment

Jason J. Saleem[1,2] and Dustin T. Weiler[1,3]

[1] Department of Industrial Engineering, University of Louisville, Louisville, KY, United States of America
[2] Center for Ergonomics, University of Louisville, Louisville, KY, United States of America
[3] Department of Industrial and Systems Engineering, University of Wisconsin-Madison, Madison, WI, United States of America

## ABSTRACT

Potential benefits of multiscreen and multiple device environments were assessed using three different computing environments. A single factor, within-subject study was conducted with 18 engineering students in a laboratory experiment. Three levels for the computing environment factor included one with a desktop computer with a single monitor (control, condition A); one with a desktop with dual monitors, as well as a single tablet computer (condition B); and one with a desktop with a single monitor, as well as two tablet computers (condition C). There was no statistically significant difference in efficiency or workload when completing scenarios for the three computing environments. However, a dual monitor desktop with a single tablet computer (B) was the ideal computing environment for the information-rich engineering problem given to participants, supported by significantly fewer errors compared to condition C and significantly higher usability ratings compared to conditions A and C. A single desktop monitor with two tablet computers (C) did not provide any advantage compared to a single desktop monitor (A).

# INTRODUCTION

As having more than one computing device and/or monitors is becoming more feasible for individuals, a future trend is the of adoption of a multiscreen and multiple device approach to cope with distractions and multiple tasks. Although this may seem counterintuitive, more screens and possibly more devices may help focus one's attention rather than serve as a distraction, making multiple tasks viewable at a glance across multiple screens and devices (*Thompson, 2014*). Assuming each device has a different primary purpose, the additional screens may begin to approximate some of the inherent affordances of paper. That is, spreading out papers on a desk lets one's eyes easily scan, which is a property hard to replicate on a single computer screen. Thus, coordination of multiple computing devices and screens is a strategy that may potentially improve one's performance in an information-rich environment by focusing their attention and reducing their mental workload. Combining multiple screens and information devices has recently been studied qualitatively, in the field (*Jokela, Ojala & Olsson, 2015*). However, little quantitative experimentation has been

Corresponding author
Jason J. Saleem,
jason.saleem@louisville.edu

done as to how a multi-device setup might affect task performance, which is the main objective of this study.

The study described in this paper is a natural evolution of a previous study that involved paper-based workarounds to using the electronic health record (EHR) (*Saleem et al., 2009*). In that study, we found that paper served as an important tool and assisted healthcare employees in their work. In other cases, paper use circumvented the intended EHR design, introduced potential gaps in documentation, and generated possible paths to medical error. Investigating these paper processes helped us understand how the current exam room computing and EHR were not meeting the needs of the clinicians. The "forgotten" power of paper, including its ability to serve as a reliable cognitive memory aid and to help focus attention on important information, were lost as EHRs began to take shape. Today, a multiscreen and multiple device work environment has become a trend. But, how to optimize the use and coordination of these multiple screens and devices is not known. This type of environment may help simulate the forgotten power of paper by replicating many of the lost affordances of paper-based processes, such as easy visual attention switches across screens, as well as the display of the most important information, separated by function or purpose across screens and devices. The objective of our study was to understand how to optimize this type of multiscreen and multiple device environment for improved user performance and satisfaction, and reduced mental workload. The adoption of a multiscreen/multiple device ecosystem is one intriguing potential solution to reduce the need for paper-based workarounds to a single computer system.

Other researchers are investigating computer-based multi-display environments. *Carter, Nansen & Gibbs (2014)* introduced the phrase 'contemporary screen ecologies' to refer to multiple screen configurations, where the modern computing experience is becoming complex as laptops, smartphones, and tablets enter the personal computing space. They demonstrate how multiple screens have transformed the way digital games are played and experienced. In addition to gaming, others have studied additional screens in relation to television (TV) viewing (e.g., *Brown et al., 2014*; *Neate, Jones & Evans, 2015*). Laptops, smartphones, and tablet computers are often used as a second screen while watching TV, which has implications for designing companion content for the secondary screen, as well as attention switches between the screens and posture when the individual is engaged with each screen (*Brown et al., 2014*). *Vatavu & Mancas (2014)* studied visual attention patters for multiscreen TV and noted that more screens may demand higher cognitive load and increased visual attention distributed across displays in the context of watching TV programming. *Grubert, Kranz & Quigley (2016)* have identified a number of design challenges for multi-device ecosystems, including those related to single user interaction, which is the type of interaction explored in the current study. These challenges include varying device characteristics, fidelity (quality of output and input characteristics), spatial reference frame, foreground-background interaction, visibility and tangibility (*Grubert, Kranz & Quigley, 2016*). In addition, there are technological challenges that should be considered in a multi-device ecosystem. One technological challenge that is especially relevant to the current study is 'heterogeneity of software platforms and form factors' (*Grubert, Kranz & Quigley, 2016*). By including a desktop and tablet(s) that use different

operating systems in our multi-device ecosystem, there is a potential for inconsistencies with how users may interact with information across the form factors and operating systems.

There exists a large body of human–computer interaction (HCI) literature on the use of multiple screens, screen sizes, and form factors (e.g., desktop, tablet, smartphone). Previous studies in academic (*Anderson et al., 2004*; *Russell & Wong, 2005*) and hospital (*Poder, Godbout & Bellemare, 2011*) settings have demonstrated that performance is improved with the use of two monitors compared to one. For example, participants were quicker on tasks, did the work faster, and performed more work with fewer errors in multiscreen (dual screen) configurations than with a single screen (*Anderson et al., 2004*). Another study demonstrated that users do not tend to treat a second monitor as additional space. That is, participants rarely reported straddling a single window across two monitors. This is consistent with the physical gaps that are often left between monitors. Instead, users typically maximize a design to fill one monitor entirely, leaving the other monitor free for other uses (*Grudin, 2001*). Visual and physical separation between displays requires that users perform visual attention switches (*Rashid, Nacenta & Quigley, 2012*), likely increasing users' cognitive load. In one study, the authors utilized a divided attention paradigm to explore the effects of visual separation and physical discontinuities when distributing information across multiple displays. Results showed reliable detrimental effects (about a 10% performance decrement) when information is separated within the visual field, but only when coupled with an offset in depth (*Tan & Czerwinski, 2003*).

The optimal monitor size and position has also been studied. One study compared 15-, 17-, 19-, and 21-inch monitors and found that while participants' performance was most efficient with the 21-inch monitor for Excel and Word tasks, users significantly preferred the 19-inch monitor (*Simmons, 2001*). The majority (65%) of participants noted that the 21-inch monitor was too large or bulky for the average workspace (*Simmons & Manahan, 1999*; *Simmons, 2001*), although this perception may be due to the novelty of larger screens at that time. A limitation of this study was that screen resolution was not controlled for across the four screen sizes. Although there has also been experimentation with very large displays (e.g., 42-inch monitor), there are several usability issues that are barriers to adopting larger displays, including: losing track of the cursor, distal access to information, window management problems (e.g., windows pop up in unexpected places), task management problems, configuration problems, and failure to leverage the periphery (*Czerwinski et al., 2006*). Therefore, separate smaller displays (e.g., 19-inch) seems to be advantageous as compared to a single, very large display. In terms of user-preferred position of computer monitors, one study found that participants placed larger displays farther and lower while maintaining the display top at or near eye height (*Shin & Hegde, 2010*). Preferred position of the dual displays in landscape arrangement did not differ from that of a single display. Participants determined a preferred position of the multiple displays not by the overall horizontal dimension (or viewable area) of multiple displays but by the vertical dimension of the overall viewable area of a single display (*Shin & Hegde, 2010*).

In addition to multiple monitors, handheld computers such as tablets and smartphones are becoming much more accessible in the workplace. For example, in clinical care settings,

one research team noted that by making the most useful and appropriate data available on multiple devices and by facilitating the visual attention switching between those devices, staff members can efficiently integrate them in their workflow, allowing for faster and more accurate decisions (*De Backere et al., 2015*). Research on the dependence of performance on the form factor of handheld computers revealed a significant difference in completion times between the tablet and smartphone screen sizes (17.8 vs. 7.1 cm), but no differences in errors or subjectively assessed cognitive workload (*Byrd & Caldwell, 2011*). These previous studies were useful for understanding how to blend a multiple monitor environment with additional devices, such as tablet computers, for creating multiscreen environments to compare in our study. Although combining multiple screens and information devices has been studied qualitatively in the field, little quantitative experimentation has been done as to how a multi-device setup might affect task performance, which is the main objective of this laboratory study.

## METHODS

### Study design

This research was approved by the Institutional Review Board (IRB) at the University of Louisville (IRB # 16.0025). Informed consent was obtained from each participant. The study was conducted in the Center for Ergonomics lab space at the University of Louisville to test the three different computing work areas with 18 engineering students. We used a counterbalanced, within-subject design, with 'Computing Environment' as the single independent variable. The three levels of Computing Environment are shown in Fig. 1. The presentation order of the three work area computing conditions were counterbalanced across the 18 participants to control for a potential carry over learning effect. Condition A had a single desktop computer with a 19-inch monitor (baseline condition). Condition B had a desktop with dual 19-inch monitors, as well as a single tablet computer with a 9.7-inch display. Condition C had a desktop with a 19-inch monitor, as well as two tablet computers, with 9.7 inch displays. The 19-inch monitors were in fixed positions; however, the tablet computers were not fixed or propped up and could be moved based on users' preferences. A standard keyboard and mouse were used as the input devices for the monitors. The desktop had a Windows 7 operating system and the tablets were iPad Air 2′s with the iOS 10 operating system. The input for the iPads were via touch screen and electronic keyboard (no external input devices were connected to the iPads). The same resolution (1,920 × 1,080 pixels) for the 19-inch monitors was used for each condition. The resolution of the iPads was 1,536 × 2,048 pixels. These three conditions were chosen based on a review of the literature to begin to understand how a multiscreen work area may affect performance and satisfaction in an information-rich environment. We decided to include no more than three conditions for the experimental design in order to complete the entire experimental session into 1.5 h or less. Ideally, we would have varied less factors simultaneously but felt it necessary to include no more than three conditions. Therefore, we included a baseline condition (A) and two multiscreen conditions (B and C). We felt it important to include three screens for the multiscreen conditions, as there is already

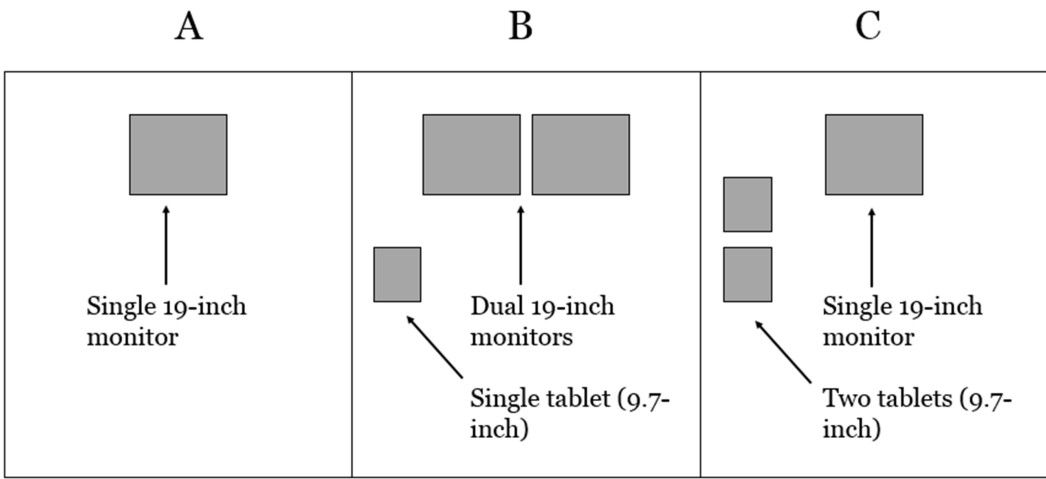

**Figure 1 Three experimental conditions for Computing Environment.** (A) Condition A had a single desktop computer with a 19-inch monitor (baseline condition). (B) Condition B had a desktop with dual 19-inch monitors, as well as a single tablet computer with a 9.7-inch display. (C) Condition C had a desktop with a 19-inch monitor, as well as two tablet computers, with 9.7 inch displays.

ample research on dual screen systems, and the inclusion of tablet computers adds further novelty, by introducing a mobile form factor to the multiscreen ecosystem.

A previous study found that a 19-inch monitor is the optimal screen size based on performance and preference (*Simmons & Manahan, 1999*; *Simmons, 2001*). Therefore, a single 19-inch monitor work area served as a baseline condition A for comparison with the multiscreen conditions. Several studies have found increased performance for dual-screen users (*Anderson et al., 2004*; *Poder, Godbout & Bellemare, 2011*; *Russell & Wong, 2005*); a dual screen set up is part of condition. The dual-screens were fixed on the horizontal plane from the user's perspective since varying screen position by depth was found to result in a performance decrement (*Tan & Czerwinski, 2003*). Although pervious research supports the use of dual-screen monitors, it is not known how performance and satisfaction is impacted with the availability of additional screens from the use of mobile technologies. Tablet computers were introduced in conditions B and C rather than other form factors such as smartphones since previous research demonstrated a significant difference in task completion times between the tablet and smartphone screen sizes (*Byrd & Caldwell, 2011*). One tablet computer was introduced in condition B to use in conjunction with the dual monitor desktop and two tablets computers were introduced in condition C to use in conjunction with a single monitor desktop. Conditions B and C incorporated multiple screens across multiple form factors (desktop monitor and tablet computer) to understand if multiple screens can help focus (or distract) users' attention in an information-rich environment (*Thompson, 2014*).

## Participants

For this study, 18 industrial engineering students (11 males, seven females) participated between March–June 2016. Industrial engineering students were chosen based on the

flow-charting tasks involved in the session; all students, except for one, had previously learned how to use a process flow chart from an undergraduate course on work design. The one exception was a graduate student with a mathematics undergraduate background. However, she was given an overview of process flow charting technique prior to data collection. Participants were between the ages of 19 and 26 years old; the median age was 23. All participants, with the exception of one, reported little or no previous knowledge of race car driving, which was the application area for the experimental tasks. One participant had a great deal of knowledge about race car driving. Ten of the participants currently used a dual-monitor set-up for their personal workstations and all but one participant had experience using tablet computers or '2 in 1' computers (tablets that convert to a laptop). Only one participant reported regularly using an iPad, which were the tablets used as part of this study.

## Dependent measures

We used performance (efficiency and accuracy), workload, and usability as measures to demonstrate improved work area computing. Specifically, improved efficiency (time to complete tasks) and accuracy (reduction of errors) using a certain work area computing condition (A, B, or C), would suggest that the work area computing set-up better supports the users' ability to efficiently and effectively complete information-rich tasks. Similarly, through improved work area computing set-up, a decrease in mental workload (and thus required attentional resources) was predicted, as measured by the NASA Task Load Index (TLX) (*Hart & Staveland, 1998*), which is commonly used to measure mental workload of a range of tasks of varying duration. We used unweighted TLX scores as the TLX dimensional weighting procedure has been found to be of limited benefit (*Hendy, Hamilton & Landry, 1993*; *Nygren, 1991*). Finally, an improved work area computing set-up would be expected to score higher on a validated usability survey; we used the Computer Usability Satisfaction Questionnaire (CSUQ) (*Lewis, 1995*). Each of these measures was used to compare the three experimental conditions for work area computing.

## Scenarios and tasks

Participants were asked to use flow process charts to document the steps that members of a National Association for Stock Car Auto Racing (NASCAR) team perform during a pit stop. Participants documented different members of the pit crew for each of the three work area computing conditions A, B, and C. The multiscreen/device conditions B and C can be described as ''related parallel use'' conditions (*Jokela, Ojala & Olsson, 2015*), where participants work on completing a single task using more than one device in parallel. The three members of the pit crew for this experiment were front tire carrier, rear tire carrier, and jack man. Participants first watched a demonstration/tutorial video that showed the roles of each member of the pit crew (*Interstate Batteries, 2012*). After this orientation, participants experienced each work area computing condition while completing a flow process chart to document a different pit crew member's tasks while watching an actual pit stop video (*ArmyRanger241, 2015*). Solutions for the flow process charts for each of the three roles were developed by one of the authors (D.T.W.), who possessed extensive

knowledge of NASCAR racing, prior to the first participant (Appendices A–C). These flowcharts use standard symbols to classify tasks as operation, transportation, storage, delay, and inspection steps in a process. We chose this particular pit stop scenario as an example of an information-rich task, where the use of multiple screens was potentially useful. Table 1 shows how the information was partitioned across the screens and devices for each condition.

## Experimental space

The laboratory in the Center for Ergonomics consisted of a participant room (134 sq. ft.) within a larger, main laboratory space (848 sq. ft.). The participant room and main laboratory space were connected with a door and a one-way mirror. The experimenter's station was located just outside of the participant room. Morae usability testing software (version 3.3.4; TechSmith Corporation, Okemos, MI, USA) connected the participant's computer and experiment's computer and was used to display the tasks and instructions to the participant. Morae was also used to video record the direct screen capture of the participant's interaction with the two desktop monitors. A webcam was used to record the participant's interaction with the iPads, and was synced with the Morae screen capture recording. Time to complete the scenarios was automatically captured by Morae.

## Procedure

After completing a demographics form, participants were given a brief verbal overview of the purpose of the experiment and then oriented to the experimental space. After watching the pit stop demonstration (tutorial) video, participants completed a flow process chart for a member of the pit crew with the work area computing conditions A, B, and C, (counterbalanced across participants) using the information available to them listed in Table 1. Documents and information needed to complete this task, including a blank flow process chart, were provided to the participant by the experimenter through email. After accessing these information items through email, participants could display them as they wished (split screen or toggle between windows to view one at a time) as long as the information items were partitioned across the monitors and devices as prescribed in Table 1. For all three conditions, the flow process chart was always located on Monitor 1 as completing the chart was the primary activity. All other information sources in Table 1 were supportive of completing the flow process chart. A dimension sheet of the pit stop area was provided so that participants could estimate distance for travel steps in the flow process chart.

Each of three pit crew roles (tire carrier, rear tire carrier, and jack man) were randomly assigned to the three conditions for each participant. After completing the scenarios for a given condition, participants were given the NASA TLX (computerized version) and CSUQ (paper-based version) surveys for mental workload and usability, respectively. Thus participants completed each survey a total of three times, one for each work area computing condition A, B, and C. After completing the final condition, the debrief session commenced, with the experimenter conducting a semi-structured interview to explore each participant's experiences with each condition (Appendix D). The debrief interview

**Table 1  Information partition across screens and devices.** Participants were asked to use flow process charts to document the steps that members of a race car team perform during a pit stop for each of the three experimental conditions for Computing Environment. The table shows how the information needed to complete the scenario was partitioned across the screens and devices for each condition.

| Screen/device | Condition A | Condition B | Condition C |
|---|---|---|---|
| Monitor 1 | Tutorial video | Flow process chart | Tutorial video |
| | Pit stop video | | Flow process chart |
| | Flow process chart | | Email access |
| | Email access | | |
| | Dimension sheet | | |
| Monitor 2 | N/A | Tutorial video | N/A |
| | | Email access | |
| | | Dimension sheet | |
| iPad 1 | N/A | Pit stop video | Pit stop video |
| iPad 2 | N/A | N/A | Dimension sheet |

was audio recorded by Morae. Participants received a $30 gift card at the completion of the debrief session as compensation for their time. The entire participation time was scheduled for 1.5 h for each volunteer.

## Hypotheses

We analyzed the different configurations (Fig. 1) to identify observations that could be tested by the experiment. Based on a review of the literature supporting the use of multiscreen and multi-device computing to improve performance in information-rich environments, as well as the possibility that multiple screens may help focus one's attention when the information and functions are parsed distinctly across each screen, the following were predicted:

*Hypothesis 1*: Participants will perform the scenarios in significantly less time and with significantly fewer errors with conditions B and C as compared to condition A (Fig. 1).

*Hypothesis 2*: Participants will experience significantly less mental workload when completing the scenarios with conditions B and C as compared to condition A.

*Hypothesis 3*: Participants will rate the usability of the work area computing set-up in conditions B and C significantly higher as compared to condition A.

*Hypothesis 4*: There will be no significant differences for any of the dependent variables between condition B and condition C.

## Analysis

The simulation study followed a single factor, within-subject experimental design. The single factor was 'Computing Environment', with three levels (A, B, C) depicted in Fig. 1. A one-way repeated measures analysis of variance (ANOVA) was planned to test for a main effect of 'Computing Environment' on each dependent outcome measure. We planned to use non-parametric statistical testing (i.e., Friedman Test) if the normality assumption of ANOVA was violated for a dependent variable. A 0.05 level of significance was applied to all statistical tests, which is typical for this type of human factors study design. Qualitative data collected from the debrief interview session were analyzed for recurrent themes across

participants. These qualitative data were collected to help explain the quantitative results. All statistical analyses were performed using Minitab software (version 18.1; Minitab Inc., State College, PA, USA).

# RESULTS

The complete statistical analyses for this study, including ANOVA tables and post-hoc analyses are available in a supplementary analysis file. We summarize the results here.

## Performance

### Time

Mean scenario completion time, with the standard deviation in parentheses, was 596.2 s (163.4 s) for condition A, 563.0 s (213.8 s) for condition B, and 589.7 s (195.5 s) for condition C. A one-way repeated measures ANOVA did not reveal a main effect of Computing Environment on time.

### Accuracy

Solutions were used to check the accuracy of each participants, flow process charts in terms of errors made. Errors included omission errors, incorrect classification of events (e.g., operation vs. transportation), and errors involving the time or distance (for transportation items) for each event. These error counts were treated as ordinal data; 6 median errors were committed by participants when completing scenarios with condition A, five median errors with condition B, and seven median errors with condition C. A Friedman Test revealed a main effect of Computing Environment on errors, $X^2(2) = 6.78$, $p = 0.034$, unadjusted for ties. Post-hoc analysis showed that the difference between conditions B and C was the only significant difference (Wilcoxon Signed Ranks Test); participants committed 40% more errors with condition C compared to condition B.

## Workload

The NASA TLX data were not normally distributed for the overall composite score or for any of the six subscales, with the exception of mental demand. Therefore, we used non-parametric testing to analyze the workload data. The Friedman Test was used to analyze the overall score and subscales and found no statistically significant differences in workload across the three conditions. A summary of the NASA TLX scores is presented in Table 2.

## Usability

The CSUQ was analyzed along an overall score and three subscales, shown in Table 3. Item 9 related to error messages and was excluded from the analysis since there were no error messages presented to participants as part of the study scenario. A copy of the complete CSUQ survey is available in Appendix E. We used one-way repeated measures ANOVA to test for a main effect of Computing Environment on the system usefulness and information quality subscales. However, the data for overall satisfaction and interface quality failed the normality assumption and so we treated those data as ordinal and used the Friedman Test for those two subscales. Statistically significant results were found for overall satisfaction,

**Table 2 Summary of NASA TLX scores (mean, standard deviation).** The table shows workload ratings for each of the six subscales and overall composite score for the NASA TLX.

| Cond | MD | PD | TD | Perf. | Effort | Frust. | Total |
|------|-----------|-----------|-----------|-----------|-----------|-----------|-----------|
| A | 54.2 (22.1) | 21.1 (14.8) | 42.2 (17.8) | 37.8 (22.7) | 51.1 (18.2) | 31.7 (25.0) | 39.7 (13.3) |
| B | 46.7 (21.9) | 22.2 (19.7) | 39.4 (21.1) | 40.6 (28.1) | 42.5 (22.2) | 31.7 (26.2) | 37.2 (14.7) |
| C | 48.1 (18.4) | 23.6 (18.0) | 43.9 (21.7) | 41.1 (24.3) | 46.9 (20.8) | 31.7 (23.8) | 39.2 (13.5) |

**Notes.**

Cond., condition; MD, mental demand; PD, physical demand; TD, temporal demand; Perf., performance; Frust., frustration; total, total composite TLX score, unweighted.

$X^2(2) = 12.19$, $p = 0.002$, unadjusted for ties; system usefulness, $F(2, 34) = 4.27$, $p = 0.022$; information quality, $F(2, 34) = 3.78$, $p = 0.033$; and interface quality, $X^2(2) = 14.53$, $p = 0.001$, unadjusted for ties. For system usefulness, post-hoc analysis (Paired $t$-tests) revealed that the significant differences are isolated between conditions A and B as well as between B and C; participants rated system usefulness 18% higher for B than A and 14% higher for B than C. Condition C is not considered different than A. For information quality, post-hoc analysis (Paired $t$-tests) revealed that the significant difference is isolated between conditions A and B; participants rated information quality 10% higher for B than A. Condition C is not considered different than A or B. For both overall satisfaction and interface quality, post-hoc analysis (Wilcoxon Signed Rank Test) revealed that B is significantly different from A and C. Participants rated overall satisfaction 16% higher for B than A and 9% higher for B than C. Participants rather interface quality 26% higher for B than A and 16% higher for B than C. However, A and C are not considered different for overall satisfaction or interface quality.

## Qualitative results

During the debrief interview, 15 of 18 participants expressed a clear preference for the computing environment in condition B (dual monitors and one iPad); three participants expressed a clear preference for condition A (single monitor); no participants expressed a preference for condition C (single monitor and two iPads). Of the 15 participants who chose the layout in condition B as best, six of them explicitly stated that the iPad was unnecessary. Conversely, three of the 15 participants expressed a clear preference for the iPad in addition to the dual monitors. When asked about an "optimal" computing environment for their work (i.e., not restricted to choosing one of the three conditions), 16 participants indicated they would prefer two desktop monitors and did not foresee a useful role provided by an additional tablet computer. One participant would prefer a single desktop monitor. And one participant indicated, "the more monitors the better". Within these 18 responses, five participants expressed a desire or noted an advantage for having a mobile device in addition to fixed monitors for portability of information (three mentioned tablet computers, one mentioned a smartphone, and one mentioned a laptop).

**Table 3   Usability scores from the Computer System Usability Questionnaire (CSUQ).** The table shows the usability ratings from the CSUQ.

| Score | Condition A | Condition B | Condition C | *p* value |
|---|---|---|---|---|
| Overall satisfaction (items 1–19) | 5.0 (1.0) | 5.8 (1.0) | 5.3 (1.0) | 0.002[*] |
| System usefulness (items 1–8) | 5.0 (1.1) | 5.9 (1.0) | 5.2 (1.0) | 0.022[*] |
| Information quality (items 10–15) | 5.1 (1.0) | 5.6 (1.0) | 5.5 (0.9) | 0.033[*] |
| Interface quality (items 16–18) | 4.7 (1.5) | 5.9 (1.1) | 5.1 (1.3) | 0.001[*] |

**Notes.**

Item 9 was excluded from the analysis as not applicable. Ratings are derived from 7-point Likert-type scales ranging from 1 = strongly disagree to 7 = strongly agree.

[*]*p* values indicate statistically significant findings ($p < 0.05$). *p* values reported for system usefulness and information quality are from a repeated measures analysis of variance (ANOVA). *p* values reported for overall satisfaction and interface quality are from the Friedman Test, unadjusted for ties.

## DISCUSSION

The results of this investigation into the benefit of using multiple screens and multiple devices were mixed; some of our hypotheses were not supported and others were partially supported. Our first hypothesis was that participants would perform the scenarios in significantly less time and with significantly fewer errors with conditions B and C as compared to condition A. While participants, on average, completed scenarios in less time with condition B, there was no statistically significant difference in time to complete scenarios for the three computing environments. One statistically significant result for errors was isolated between conditions B and C; participants committed significantly less errors when using condition B compared to C. These results suggest marginal support for our first hypothesis, but only for condition B. Condition C was not considered different than the baseline condition A for time and errors.

Our second hypothesis was that participants would experience significantly less mental workload when completing the scenarios with conditions B and C as compared to condition A. This hypothesis was not supported. There was no statistically significant difference in the NASA TLX scores when completing scenarios for the three computing environments. However, it is worth noting that condition B was scored, on average, as better than the other conditions especially on the 'mental demand' and 'effort' subscales.

The third hypothesis was that participants would rate the usability of the work area computing set-up in conditions B and C significantly higher as compared to condition A. This hypothesis was partially supported. Condition B was scored significantly higher for overall usability, system usefulness, and interface quality compared to both conditions A and C; as well as significantly higher for information quality compared to condition A. However, condition C was not scored significantly higher for any of the CSUQ scales compared to the baseline condition A.

Our final hypothesis was that there would be no significant differences for any of the dependent variables between condition B and condition C. This hypothesis was not supported. Participants committed significantly fewer errors with condition B compared to condition C. They also rated the overall usability, system usefulness, and interface quality as significantly better for the computing environment condition B compared to condition C.

## Key findings

A dual monitor desktop with a single tablet computer (condition B) was the ideal computing environment for the "information-rich" engineering problem given to participants. This is supported by converging evidence from the dependent measures as well as the qualitative debrief interviews. A single desktop monitor with two tablet computers (condition C) did not provide any advantage compared to a single desktop monitor (condition A). Although the tablet computers could be moved based on users' preferences, most participants did not re-arrange the locations of the tablets. Overall, these findings provide only marginal support for the concept we set out to investigate, which was the notion that more screens and possibly more devices may help focus one's attention rather than serve as a distraction, making multiple tasks viewable at a glance across multiple device screens (*Thompson, 2014*). The finding of a performance and usability advantage of the dual monitors in condition B is consistent with previous studies (*Anderson et al., 2004*; *Poder, Godbout & Bellemare, 2011*; *Russell & Wong, 2005*). A key difference in our study is that we provided a tablet computer in addition to the dual monitors. However, the debrief interviews were mixed as to the usefulness of the third screen provided by the tablet; some participants thought it was not helpful whereas other did find it useful. The complete lack of performance, workload, and usability differences between condition C (single monitor and two tablet computers) and condition A (single monitor) does not support the notion that a multiscreen environment can help focus one's attention. Indeed, some participants noted that using multiple screens provided by the tablet computer(s) was distracting. Others noted that while they did not hinder their tasks, they did not help.

The chosen scenario and tasks for this study was likely contributed to these inconclusive findings. Participants were asked to use flow process charts to document the steps that members of a NASCAR team perform during a pit stop. Although this was an information-rich task, with multiple videos, documents, and email to review, participants seemed comfortable managing the information sources with less than three screens. A scenario that requires a greater degree of information management from disparate sources may be more beneficial to a multiscreen ecosystem that includes screens provided by mobile form factors.

Another important consideration is that the tablets computers in our study supported touch-screen interaction, which is a different type of interaction than afforded by a keyboard, mouse, and monitor set-up with a desktop computer. Participants in our study used the tablet computers primarily for viewing videos. If the experimental tasks were to involve operations such as zooming in/out, or other more direct manipulation tasks, then the introduction of the tablet computers may have been more impactful on our study findings. This relates to the 'heterogeneity of software platforms and form factors' technological challenge for multi-device ecosystems (*Grubert, Kranz & Quigley, 2016*). It seems that this type of heterogeneity across devices in the multi-device ecosystem has the potential to have negative impacts due to inconsistencies with how users may interact with the devices. Conversely, the heterogeneity may have positive impacts if the inherent advantages each device offers can be thoughtfully designed into the ecosystem, with consideration of the work to be done.

### Limitations and future research

Ideally, we would have varied less factors simultaneously but felt it necessary to include no more than three experimental conditions to fit within the time constraints of the study sessions. Varying fewer factors at once may have yielded more insights. Also, our study focused on engineering students completing flow process charts with a race car pit stop scenario as an example of an information-rich task, where the use of multiple screens was potentially useful. A more complex scenario or application area, with a clearer distinction for parsing certain information across screens with distinctly different purposes, may be more amenable to a multiscreen and multi-device environment. For example, a physician that needs to integrate patient data and other information from multiple functions within an EHR and other related clinical information systems may be a more appropriate scenario that we intend to investigate in a future study. The number of participants in this study is relatively small. Also, our study used Apple iPad tablets; all but one of our participants had experience using tablet computers but only one reported regularly using a iPad. In addition, one participant had a great deal of knowledge about race car driving compared to the others. Each of these limitations represent potential threats to validity of the study results and should be considered when interpreting our findings. Future research should incorporate other types of tablets and mobile devices, as well as more advanced ones that may better approximate the forgotten power of paper (e.g., *Tarun et al., 2013*).

## CONCLUSION

We designed a study to investigate the potential benefit of multiscreen and multiple device environments using three different computing environment conditions. Scenarios completed with condition B, which included a desktop with dual 19-inch monitors, as well as a single tablet computer with a 9.7-inch display, resulted in significantly less errors compared condition C, which included a desktop with a with a 19-inch monitor, as well as two tablet computers, with 9.7 inch displays. Condition B was also resulted in significantly higher usability ratings compared to condition C and compared to a baseline condition A (single desktop computer with a 19-inch monitor). Our findings are consistent with the literature that show better performance using a dual screen set-up. However, our findings provide only marginal support for the benefit of incorporating additional screens in the form of tablet computers during information-rich, complex tasks. Based on these results, we recommend a computing work environment with dual screen monitors, with an optional tablet computer, for complex and information-rich computing tasks.

## ACKNOWLEDGEMENTS

A portion of the results from this study was presented at the Human Factors and Ergonomics Society (HFES) International Annual Meeting, Austin, TX, October 9–13, 2017.

### Funding

This work was supported by the Department of Industrial Engineering, J.B. Speed School of Engineering, and the Center for Ergonomics, University of Louisville. The authors received no external funding for this work. The funders had no role in study design, data collection and analysis, decision to publish, or preparation of the manuscript.

### Grant Disclosures

The following grant information was disclosed by the authors:
Department of Industrial Engineering.
Speed School of Engineering.
Center for Ergonomics, University of Louisville.

### Competing Interests

The authors declare there are no competing interests.

### Author Contributions

- Jason J. Saleem conceived and designed the experiments, performed the experiments, analyzed the data, contributed reagents/materials/analysis tools, prepared figures and/or tables, performed the computation work, authored or reviewed drafts of the paper, approved the final draft.
- Dustin T. Weiler performed the experiments, analyzed the data, contributed reagents/materials/analysis tools, prepared figures and/or tables, performed the computation work, authored or reviewed drafts of the paper, approved the final draft.

### Ethics

The following information was supplied relating to ethical approvals (i.e., approving body and any reference numbers):

The University of Louisville Institutional Review Board (IRB) granted approval to carry out the study within its facilities (IRB # 16.0025).

### Data Availability

The raw data are provided in a Supplemental File.

### Supplemental Information

Supplemental information for this article can be found online at http://dx.doi.org/10.7717/peerj-cs.162#supplemental-information.

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
