# Peer review of "Performance, workload, and usability in a multiscreen, multi-device, information-rich environment"

_PeerJ Computer Science, doi:10.7717/peerj-cs.162_

## Round 0.1 · original submission · Major Revisions

In general this work has been well received by reviewers, which is entirely expected given the topic and its prior exposure as preprint and conference paper.

All reviewers remarked on the paper's quality of English, but one common criticism is that there appears to be no analysis of subjects' behavioural patterns during testing, and no attempt at qualitative analysis of the semi-structured interviews beyond anecdotal comments in the discussion. It is understandable if this was beyond the scope and available resources of the authors, but as R2 and R3 point out there may be valuable insights to be gained concerning how test subjects achieved task goals through particular layout configurations within each device setup. I would also add that an encoding and semi-quantitative analysis of pattern of behavioural cues exhibited by subjects during the procedure may also elicit further insights. A second, perhaps more significant issue is highlighted by R2 that the study design for statistical analysis of the results was not correct. R3 echoes this and requests more details and statistical support for the post-hoc results.

Major revisions.

1. Please revise the study design and present a complete statistical analysis in a reproducible form. Please cite and quote versions of any software used and provide scripts if available. If interactive stats tools are employed (e.g. Excel) it is also important to state version and platform.

2. R2 and R3 both consider the connection between this work and your previous work on Human Factors in Clinical Informatics and Electronic Health Records to be tenuous. Whilst the general topics are clearly related, this manuscript will be strengthened by addressing their requests to include more references and discussion on the topic of task efficiencies and multi-modal computing.

3. Please discuss any threats to validity of your results given that the number of subjects used in this study is small, one participant had prior domain knowledge of the test tasks, and a few others had experience with the application environments employed in one scenario.

Minor revisions and questions.
4. Please also see my questions, comments and suggested typographical revisions in the annotated manuscript.

Further work

5. Whilst all reviewers recommend qualitative analysis examination of any behavioural data that you have also collected from this experiment this may not be feasible, or allowed by your IRB. If an analysis of the interview data could also be provided, however, this would add further value to this (already valuable) work.

·

Basic reporting

no comment

Experimental design

no comment

Validity of the findings

no comment

Additional comments

This is a very well-designed, -executed, and -written study. My comments are only suggestions for improvement, not requirements for publication.

One factor that you might want to discuss is how a tablet with a touch-screen supports different types of interaction than does a keyboard, mouse, monitor set-up. If the experimental task were to involve operations such as zooming in/out, or other more direct manipulation tasks, then the impact of tablets may be different than what this study found.

Another suggestion is to restate the main objective of the study (line 60-61) at the end of the Introduction. This would help to connect the literature review to the methods.

Lastly - Are there any qualitative results on how the participants used the displays/tools in completion of the task? Did they generally arrange the windows the same way, or was there a lot of variation (within the constraints presented in table 1)? I realize qualitative analysis was not the main objective of the study, but it is possible it may help explain the performance results.

Reviewer 2 ·

Basic reporting

Interesting perspective, with a useful research question on multiple interfaces for accessing complex information. The reference to medical information use and electronic health records (EHR) seems awkward and disconnected; although the EHR context was previously studied by the authors, the current task (NASCAR pit stops) and analysis flow charting is not directly related to the EHR environment. The raw data and tables are shared, but the statistical analyses are not.

Experimental design

The research question is well defined, and the task analysis is based on an interesting approach. However, the sample is constrained to undergraduate students in a single domain of engineering, and is a small sample (only 18 total). This strongly limits generalizability of the design to a particular set of tasks and user populations. If the tablets were free to be moved, did the authors actually measure their location when the study participants moved them? That would be helpful from a perspective of learning how much variation in configuration is desired / expected among a general population of users.

Validity of the findings

The data analysis section seems incomplete, especially compared to the capability of sharing raw data (and thus statistical analysis code or F-ratio tables). The design of the study is actually a repeated-measures, within-subjects design, but it is unclear if the analyses actually used this design rather than the more common (but incorrect) between-subjects, independent groups design. (More clear and evident presentation of the ANOVA models would clarify this question. This lack of clarity / possible confusion in analysis is the primary objection I have to publishing the paper in its current form.

Additional comments

There are a number of tasks for which this study design is relevant and potentially helpful for user experience / usability / workload analysis. It is also much appreciated that the authors distinguish monitor size from screen resolution (the largest monitors from 2000 would have much lower resolution, and thus poorer readability, than current monitors with much higher resolution). However, the reference to EHR tasks seems little more than a chance for the authors to cite their own prior work; there is no shared context (or even explanation of EHR tasks) that seems to link those references to the current study application. Even with that concern, the bigger critique (from a revise and resubmit standpoint) is the lack of clarity, shared information, or explanation of the analysis of variance model between within subjects and independent groups designs.

·

Basic reporting

Discussion of related work needs adjustments. For details, see main review below.

Experimental design

Rationale for study design should be improved. For details, see main review below.

Validity of the findings

Statistical reporting should be improved. For details, see main review below.

Additional comments

The authors conducted a lab-based within-subjects experiment (n=18) to evaluate the effect of multi-display environments on performance, subjective workload and usability. Specifically, following three environments were compared: desktop computer with single screen (C1), desktop computer with dual screen and additional tablet (C2), desktop computer with single screen and two additional tablets (C3). No statistically significant differences in efficiency or workload were indicated. C2 did result in statistically significant higher usability ratings compared to C1 and C3 and to significantly fewer errors compared to C3.

The topic is timely and relevant. The paper is written in an understandable manner.
The introduction is brief (1 page), out which half a page refers to a previous paper-based study by the authors themselves. This brief introduction should be expanded and put into context of actual computer-based multi-display environments.

Some (mostly older) related work is listed but the authors miss to discuss how their work differentiates from previous work. I would suggest considering discussing articles and studies about recent multi-display environments (also including second-screen applications in TV and gaming settings) e.g.:

Brown, A., Evans, M., Jay, C., Glancy, M., Jones, R., & Harper, S. (2014, April). HCI over multiple screens. In CHI'14 Extended abstracts on human factors in computing systems (pp. 665-674). ACM.

Carter, M., Nansen, B., & Gibbs, M. R. (2014, October). Screen ecologies, multi-gaming and designing for different registers of engagement. In Proceedings of the first ACM SIGCHI annual symposium on Computer-human interaction in play (pp. 37-46). ACM.

Grubert, J., Kranz, M., & Quigley, A. (2016). Challenges in mobile multi-device ecosystems. mUX: The Journal of Mobile User Experience, 5(1), 5.

Neate, T., Jones, M., & Evans, M. (2015, April). Mediating attention for second screen companion content. In Proceedings of the 33rd Annual ACM Conference on Human Factors in Computing Systems (pp. 3103-3106). ACM.

Vatavu, R. D., & Mancas, M. (2014, June). Visual attention measures for multi-screen TV. In Proceedings of the ACM International Conference on Interactive Experiences for TV and Online Video (pp. 111-118). ACM.

Section 2.1 introduces the study design. While some rationale is presented on why the study design was chosen as it was, the authors could add a clear description on the limitations and interdependences of their chosen setup (i.e. multiple factors such as screen type, position and size were varied simultaneously). Otherwise, the experimental description follows scientific standards.

The presentation of results should be improved by including effect sizes and statistical power analysis, given an experiment with 18 * 3 = 54 samples for three conditions. Also, as the authors highlight statistically significant difference for the usability ratings they should report the statistics for the post-hoc comparisons (section 3.3).

Section 3.4 presents some brief qualitative findings, with one being that participants would prefer a dual stationary monitor setup, without further highlighting the role of the additional tablet in C2.

Section 4. (Discussion) seems not like an actual discussion but solely a summary of how the results aligned with or differed from the hypotheses.

Section 4.1 (Key findings) summarizes the inconclusive results of the experiment. Beyond what is discussed in the paper I would have liked to see a discussion about the potential impact of the chosen task on the results.

Again, my hypothesis is, that if the authors would have varied fewer factors at once, the experiment might have resulted in more insights compared to the experiment conducted.

To summarize, the authors tackle an important and timely topic. The paper is overall well written but suffers from a suboptimal differentiation from prior work and a study design with (potentially too) many factors that were varied at once. Still, after addressing the issues (that can be fixed) raised in this review, the paper can add value to the existing literature.

---

## Round 0.2 · accepted · Accept

Many thanks for considering and responding to reviewer comments on your previous draft, and providing a revised manuscript and statistical analysis. As Reviewer 2 notes - the improved rigour and additional commentary regarding behavioural observations of study subjects adds weight to your study's findings, and the new section specifically addressing threats to validity will help to address any concerns that readers may have regarding study design and methodology.

I have provided an annotated PDF which includes corrections for grammar and typos, and highlights text that should be reworded for clarity in the final version.

# Reviewer 2 ·

Basic reporting

No comment

Experimental design

Improved from prior design, now achieves "Pass"

Validity of the findings

It is still questionable whether findings from 1995-2005 with monitor resolutions of the era can be directly compared to more recent findings, but overall, the authors' edited explanations and conclusions are significantly improved.

Additional comments

The authors have done a very competent job of reviewing and responding to the reviewers' comments, and conducting a suitable revision of the statistical analysis. The additional commentary that most participants did not adjust the positions of the monitors is also useful in supporting the comments that additional monitors and functionality did not significantly improve participant performance or workload.